# Solid State Fermentation as a Tool to Stabilize and Improve Nutritive Value of Fruit and Vegetable Discards: Effect on Nutritional Composition, In Vitro Ruminal Fermentation and Organic Matter Digestibility

**DOI:** 10.3390/ani11061653

**Published:** 2021-06-02

**Authors:** Jone Ibarruri, Idoia Goiri, Marta Cebrián, Aser García-Rodríguez

**Affiliations:** 1AZTI, Food Research, Basque Research and Technology Alliance (BRTA), Parque Tecnológico de Bizkaia, Astondo Bidea, Edificio 609, 48160 Derio, Spain; mcebrian@azti.es; 2NEIKER—Basque Institute for Agricultural Research and Development, Basque Research and Technology Alliance (BRTA), Campus Agroalimentario de Arkaute s/n, 01192 Arkaute, Spain; igoiri@neiker.eus (I.G.); aserg@neiker.eus (A.G.-R.)

**Keywords:** circular economy, efficiency, by-products, *Rhizopus*, alternative protein, ruminant feed

## Abstract

**Simple Summary:**

A huge quantity of fruits and vegetables are wasted every year, having a negative impact in both the economy and the environment. Valorizing them as animals’ feeds would contribute to reduce feeding cost and, at the same time, would be in the interest of prevention of resource wastage and better economy of the processing plants. The aim of this study was, on the one side, to transform fruit and vegetable discards using solid state fermentation (SSF) to a stabilized product enriched in protein and, on the other side, to evaluate its suitability for ruminants feeding by determining the in vitro organic matter digestibility, fermentation characteristics and methane production of the control and the fermented product. As a result, it was found that SSF reduced the organic matter and reducing sugar content of the fermented product, while crude protein and fiber fractions were increased. In conclusion, SSF led to a stabilized feed ingredient enriched in protein, but at the expense of digestibility reduction.

**Abstract:**

This research aimed to evaluate in vitro organic matter digestibility, fermentation characteristics and methane production of fruit and vegetable discards processed by solid state fermentation (SSF) by *Rhizopus* sp. Mixtures were composed of approximately 28% citric fruits, 35% other fruits and 37% vegetables. Fruit and vegetables were processed and fermented to obtain a stabilized product. Nutritional characterization and in vitro ruminal fermentation tests were performed to determine the effect of fungal bioconversion on digestibility, end products and gas production kinetics. Results indicate that SSF reduced organic matter and reducing sugars, while it increased crude protein and neutral detergent fiber, acid detergent fiber and neutral detergent insoluble protein. The in vitro gas production showed that SSF led to a reduction of the organic matter digestibility (*p* < 0.001), short chain fatty acids (SCFA; *p* = 0.003) and CH_4_ (*p* = 0.002). SSF reduced the gas production from the insoluble fraction (*p* = 0.001), without modifying the production rate (*p* = 0.676) or the lag time (*p* = 0.574). Regarding SCFA profile, SSF increased acetic (*p* = 0.020) and decreased propionic (*p* = 0.004) and butyric (*p* = 0.006) acids proportions, increasing acetic to propionic (*p* = 0.008) and acetic plus butyric to propionic (*p* = 0.011) ratios. SSF succeeded in obtaining a stabilized material enriched in protein, but at the expense of a reduction of protein availability and organic matter digestibility. These changes should be considered before including them in a ruminant’s rations.

## 1. Introduction

One of the main objectives set by the European Union (EU) is obtaining a sustainable food chain where reducing food waste and producing new sustainable food and feed sources are two key aspects. It is estimated that around 20–30% of produced food in the EU is wasted, while 5 million tons of raw food is used in animal feed with an upward trend to 7 million tons for 2025 [1]. In addition, the demand for animal feed is increasing in addition to the need for new protein sources. Imported soybean meal is the main protein source used in animal feeding in the EU, which increases the environmental impact on the sector.

On the other hand, the feed sector is being challenged because of an increase in the demand for livestock animal feed; coupled with land, soil and water shortage; competition between crops for food or biofuel production and climate change. In this context, one of the keys to developing a sustainable livestock system is the reduction of food waste, such as including it as new resources for animal feed that do not compete with human food.

Fruit and vegetables are the greatest fraction of wasted food [2,3], generating more than 12 million tons of discards per year in Europe [4]. The major obstacles to their use as quality animal feed are the low nutritional value of some of these by-products (mainly due to low protein content), their high humidity and the possible presence of mycotoxins. Therefore, it is essential to implement technologies to account for these aspects, especially to improve the protein content, in a sustainable, economically feasible and controlled way [5,6].

In this context, the use of solid state fermentation (SSF) to enrich fruit and vegetable discards to obtain an alternative ingredient for feed manufacturers could be an exceptional example of resource efficiency in the circular economy of the EU. Previous studies on SSF of several food industry by-products, including fruit and vegetables, observed important increases in the protein content [7,8], while releasing phenolic compounds with antioxidant activity [9,10,11]. Among fungi, *Rhizopus* is one of the most interesting fungal genus for SSF, due to its simple nutritional requirements and growing conditions [12] and the high variety of enzymes that it can produce [13]. In addition, SSF is described as a simple treatment for digestibility improvement [14,15]. It has been successfully applied for ruminal digestibility improvement in fiber rich by-products, like corn stovers [16,17,18], wheat straw [19], camelia seed [20] and corn straw [21], among others. During the fermentation of these substrates, fungi improve fermented products’ digestibility due to their ability to release several enzymes (amylases, xylases, celullases), which degrade plant cell walls [22,23], allowing the rumen microorganisms to access the polysaccharides [18].

On the contrary, few studies have been carried out about the effect of SSF on ruminal digestibility of fruit and vegetables discards, with most of them being focused only on the protein content increase and bioactive compounds release [8,11]. The lack of knowledge about the effect of fermented fruit and vegetables on ruminal fermentation and diet digestibility limit their use in ruminant rations. In this sense, identifying digestibility of the fermented product could add extra information about the applicability of the final product, which is necessary to maximize the effective use of these fermented by-products. Therefore, the present study aimed to explore the effects of fungal SSF of fruit and vegetable discards on their nutrient profile and on in vitro digestibility and fermentation kinetics, with a view towards including them as a suitable product for ruminant feed.

## 2. Materials and Methods

The study comprised two experiments. The first experiment consisted of a short-term in vitro batch fermentation trial designed to assess the effect of the SSF on rumen fermentation of fruit and vegetable mix discards. The second experiment consisted of a long-term in vitro batch fermentation trial designed to study differences in fermentation kinetics.

### 2.1. Microorganism, Culture Media and Inoculum

The *Rhizopus* strain (ROR004, internal code) was isolated and characterized in our laboratory [7] to be used as a fermentation agent. 

Potato dextrose agar (PDA) and buffered peptone water (both from Oxoid, Basingstoke, Hampshire, UK) were used for fungal propagation, count and dilution when required. Tween 80 (Merck, Darmstadt, Germany) was used for inoculum preparation. All media were prepared as recommended by the producer and sterilized at 121 °C for 15 min. Plates for total fungal counts were incubated at 30 °C for 48 h.

Spore suspension was prepared with the mycelia formed in a PDA plate after 5 days of incubation at 30 °C. After collecting the mycelia and mixing it with 20 mL of sterile distilled water (0.01% Tween 80), suspension was maintained for 24 h at 30 °C and filtered thought 300 µm sterile filter to obtain the liquid spore suspension.

### 2.2. Fruit and Vegetable Discards

Substrates for SSF were three independent fruit and vegetable discard mixtures obtained from Mercabilbao (commercial perishable food distribution center, Basauri, Spain) to achieve true replications [24,25]. The mixtures were composed of approximately 28% citric fruits (tangerine, orange and lemon), 35% other fruits (nectarine, apple, pear, watermelon, pomegranate and banana) and 37% vegetables (tomato, pumpkin, onion, green bean, pepper, leek, artichoke, cabbage, carrot, broccoli, potato, asparagus, chard and lettuce). After cutting (R 10 v.v, Robot coupe, Mataró, Spain), the samples were centrifuged in a vertical axis centrifuge with filter bag (Comteifa, Badalona, Spain) in order to remove water excess, and each sample was divided into 2 subsamples, the control (CTR, without treatment) and the one that will be subjected to SSF: the fermented sample. Both subsamples were dried at 60 °C for 2 h and sterilized (121 °C, 15 min).

### 2.3. Solid State Fermentation 

SSF was carried out in three independent processing runs at 30 °C for 192 h on a 1680 cm^2^ surface plastic tray, with an estimated substrate density around 0.09 g of dry substrate/cm^2^. Plastic trays were cleaned with ethanol 70% and exposed to UV light for 30 min before adding the substrate. SSF runs of FVS were inoculated with 10^7^ cfu/g. Plastic trays were covered with lids but without closing them hermetically to enable the air entrance. Trays were weighed before and after the SSF process. 

### 2.4. Short-Term In Vitro Batch Fermentation Trial

Samples were dried at 60 °C in a forced-air oven for 48 h (SELECTA, Barcelona, Spain) and were ground to pass a 1-mm screen (RETSCH ZM-200, Llanera, Spain). The six samples served as the substrate in three in vitro runs. In each of the runs, rumen fluid was collected from one multiparous Latxa ewe slaughtered for production purposes. Before slaughtering, ewes were fed a basal diet (80% meadow hay and 20% compound feed) for 3 weeks and had free access to fresh water and feed. Ruminal fluid was collected before the morning feeding and strained through four layers of cheesecloth into a pre-warmed thermos flask.

Approximately 500 mg of CTR or SSF sample of the three independent processing runs were weighed into 125-mL serum bottles, 50 mL of culture fluid was added (1:4 ruminal fluid and phosphate–bicarbonate buffer, respectively; [26]) and bottles were crimp sealed. Each sample was incubated in triplicate, and bottles were incubated at a constant temperature (39 °C) in an incubator for 24 h. Gas production was released at 2, 4, 6, 8, 10, 12 and 15 h post-inoculation to avoid pressure in the bottle headspace exceeding 48 kPa, as suggested by Theodorou et al. [27]. After 24 h of incubation, bottles were put in the fridge for 15 min to stop fermentation for subsequent sampling of short chain fatty acid (SCFA) determination. 

### 2.5. Long-Term In Vitro Batch Fermentation Trial

The animals, substrates and incubation procedures were the same as those described in the previous section. Approximately 500 mg of CTR or SSF samples of the three independent processing runs were weighed into 125-mL serum bottles and incubated in an incubator at 39 °C with 10 mL strained rumen fluid and 40 mL of medium [26] to determine rate and extent of gas production by reading gas pressure in the bottle headspace at 2, 4, 6, 8, 10, 12, 15, 24, 30, 36, 48, 72 and 96 h post-inoculation, using a semi-automated pressure transducer following the technique proposed by Theodorou et al. [28] and modified by Mauricio et al. [29]. Pressure values, corrected for the quantity of substrate dry matter (DM) incubated and gas released from blanks, were used to generate gas volume estimates.

### 2.6. Chemical Analyses

CTR and SSF samples of the three independent processing runs were dried in a forced-air oven (SELECTA, Barcelona, Spain) and were ground to pass a 1-mm screen (RETSCH ZM-200, Llanera, Spain). DM content (method 934.01) was determined following [30]. Ash content was determined by ignition of the dried material (method 942.05). Nitrogen content (method 941.04) was determined using the macro-Kjeldahl procedure on a Kjeltec Auto 1030 (Foss, Hillerød, Denmark). Neutral detergent fiber (NDF) was determined with use of an alpha amylase, but without sodium sulfite, and was expressed free of ash [31]. Acid detergent fiber (ADF) was determined and expressed exclusive of residual ash [32]. Neutral detergent insoluble protein (NDICP) was determined by analyzing the NDF residues for Kjeldahl nitrogen. Total reducing sugars were determined by the Dinitrosalicylic acid (DNS) method [33] adjusted to the microplate assay procedure (Thermo Fisher Scientific, Roskilde, Denmark). Briefly, the DNS acid reagent was prepared by dissolving 8 g of NaOH in 100 mL of distilled water. Then, 5 g of DNS (Fischer Scientific, Loughborough, UK), 250 mL of distilled water and 150 g of potassium sodium tartrate tetrahydrate (Sigma-Aldrich, Steinheim, Germany) were added and made up to the volume (500 mL). Sample, blank or standard (25 μL), different concentrations of D-glucose (Fischer Scientific, Loughborough, UK), and 25 μL of DNS reagent were added to each well and incubated for 10 min at 100 °C. The microplate was rapidly cooled in an ice bath and 250 μL of distilled water was added to each well. Absorbance was read at 540 nm. 

In vitro organic matter digestibility (IVOMD) in the short term in vitro trial was calculated as described by Pell and Schofield [34], whereby 45 mL of a neutral detergent solution was added to each bottle and warmed at 105 °C for 1 h; then, the bottles were cooled, filtered through glass filter crucibles (Porosity 1) and washed with distilled water, ethanol and acetone. The remaining sample was dried at 100 °C overnight and then burned in a muffle furnace at 500 °C to obtain true IVOMD values.

The analysis of the SCFA (acetic, propionic, butyric, isobutyric, valeric and isovaleric) of rumen samples was performed by gas chromatography using a flame ionization detector. A volume of 4 mL of rumen liquor mixed with 1 mL of a solution of 20 g/L of metyl-valeric acid as an internal standard in 0.5 N HCl was centrifuged (15,000× *g* for 15 min at 4 °C) to separate the liquid phase from the feed residuals. After, the liquid phase was microfiltered (premium syringe filter regenerated cellulose, 0.45µm 4 mm, Agilent Technologies, Madrid, Spain), and 0.5 µL of liquid phase was directly injected in the apparatus (Agilent 6890 N, Agilent, Spain) using a semicapillary column (30 m × 530 um; 1-µm particle size; HP-FFAP, Agilent, Spain) kept at 300 °C in the injector with a hydrogen flow rate of 40 mL/min, air flow 400 mL/min and make up (nitrogen) 25 mL/min flow. The injection loop was 20 µL. Individual SCFA were identified using a standard solution of 4.50 g/L of acetic acid, 5.76 g/L of propionic acid, 7.02 g/L of butyric acid and isobutyric acid and 8.28 g/L of valeric acid and isovaleric acid in 0.1 N H2SO4 (A6283, P1386, B103500, I1754, 240370, 129542, respectively; Sigma-Aldrich, Madrid, Spain). Quantification expressed in mmol/L was done using an external calibration curve based on the standards described above. Data were expressed in mol/100 mol.

### 2.7. Calculations and Statistical Analysis

Stoichiometric methane values were estimated using equations proposed by Blϋmmel et al. [35] based on the stoichiometry of Wolin [36].

Fermentation kinetics were described according to the exponential model described by Krishnamoorthy et al. [37] as:(1)$$Y=A\left(1-e^{-c\left(t-L\right)}\right)$$
where Y is gas production (mL/g DM) at time t, A is gas production from the insoluble fraction (mL/g DM), c is the gas production rate constant for fraction A (h^−1^) and L is the lag time prior to gas production (h).

The parameters A, c and L for each bottle were calculated using a non-linear regression procedure, which minimizes actual distances of data points to fitted curves by Marquardt’s algorithm.

The total number of observations was 3 processing runs × 2 treatments (CTR and SSF) × 3 in vitro incubation runs × 3 lab reps. = 54; however, after averaging incubation runs and lab replicates, the remaining 6 observations were subjected to analysis of variance using the GLM procedure [38]. The statistical model, therefore, only included the fixed effect of the treatment. The least squares means for treatments are reported. Treatment means were separated using a Bonferroni adjustment, and significant effects were declared at *p* < 0.05.

## 3. Results

Table 1 shows the chemical composition of fruit and vegetable mix discards subjected to or not subjected to SSF. Solid state fermentation reduced OM content *(p <* 0.001) and reducing sugar content (*p* < 0.001) but increased CP (*p* < 0.001), NDF (*p* < 0.001), ADF (*p* < 0.001) and NDICP (*p* < 0.001) fractions.

Effects of SSF on the gas production profile of fruit and vegetable mix discards can be seen in Table 2. Solid state fermentation reduced the gas production from the insoluble fraction (*p* = 0.001), without modifying the gas production rate (*p* = 0.676) or the lag time prior to gas production (*p* = 0.574).

Effects of SSF on in vitro digestibility and fermentation parameters are shown in Table 3. In vitro organic matter digestibility was lower for the fermented substrate (*p* < 0.001). SSF reduced total SCFA (*p* = 0.003) and CH_4_ production (*p* = 0.002). However, SCFA (*p* < 0.001) and CH_4_ (*p* = 0.001) related to truly digestible substrate were increased with SSF. Solid state fermentation led to a shift in the fermentation patterns towards increased acetic (*p* = 0.020) and decreased propionic (*p* = 0.004) and butyric (*p* = 0.006) acids proportions. As a consequence, SSF increased acetic to propionic (*p* = 0.008) and acetic plus butyric to propionic (*p* = 0.011) ratios. SSF also increased proportions of isobutyric (*p* = 0.003), isovaleric (*p* = 0.003) and total BCVFA (*p* = 0.003).

## 4. Discussion

Sustainable livestock development requires novel feed resources to reduce feed costs and that do not compete with human food ingredients. Taking into account that feed is one of the largest costs in animal production, searching for economically interesting alternatives or new feedstuffs has been a hot topic in animal research in the last decades. In this context, a very interesting alternative is the use of agro-industrial residues in the animals’ rations. A large amount of vegetable by-products are wasted every year. Its disposal into the environment, being highly biodegradable, results in the production of a foul smell and affects the aquatic life and ecosystem. Utilization of such a “waste” in animals’ nutrition would contribute to reduce feeding cost and, at the same time, would be in the interest of prevention of resource wastage and better economy of the processing plants.

However, many of these residues have properties that may compromise diet digestibility or animal production, such as their inherent nutritional composition (low protein or high lignin content and fiber proportion) or the presence of toxic or anti-nutritional compounds (mycotoxins, phenolic compounds) [39].

In this sense, some authors report SSF as a promising alternative in the use of these agro-industrial by-products as a culture medium in order to account for these problems, making its use feasible in animal feed [40]. It has been also shown that SSF is one of the most suitable and economic techniques for detoxifying or enhancing protein enrichment, as well as for an efficient digestion and utilization of lignocellulosic agricultural fibrous feeds and fodder residues [14,18]. Therefore, this can enhance their feed values and bring benefits both to the economy and the environment, promoting the circular economy.

During the SSF, the fungus executes a repertory of extracellular enzymes allowing the fungus to obtain nutrients from complex polymers while simultaneously producing changes in the chemical composition of the substrate, in addition to the production of other metabolites [41].

The fruit and vegetable discard used in this trial presented a limited CP content, so one of the objectives of the SSF process was to enrich the CP content in the fermented product. The current 15.7% increase in the CP content of the fermented substrate may be due to the high production of fungal cell mass, which led to a reduction of reducing sugars, and consequently the production of protein within the fungus population. This is in agreement with other previous studies [42,43,44] that demonstrated that some fungal species were able to increase the CP level in agro-industries wastes.

Published results using substrates with similar protein content, like rice bran [41,45] and fruit and vegetable wastes [46], reported similar protein content gain (close to 1.5 fold in both cases).

In the studied case, the production of fungal cell mass during fermentation process resulted not only in CP increase but also increased NDICP contents, which could be explained by the growth of *Rhizopus* biomass. *Rhizopus* cell wall is a complex heteropolysaccharide system mainly composed by chitin and chitosan (polymers of N-acetyl-d-glucosamine attached by β-(1,4)-glycosidic linkage), mucoran, mucoric acid and glucan, whose proportion is dependent on the stage of development of the fungus [47], and could in turn explain the greater NDF and NDICP observed in the fermented substrate. These results are of practical feeding importance because NDICP is slowly degraded in the rumen and constitutes a major portion of the ruminal undegraded protein content [48]. These results agree with those of Silveira and Badiale-Furlong [49] and Ranjan et al. [50], who found a decrease in CP digestibility in these fermented products, and Nicolini et al. [51] who found a decrease in the in vitro true digestibility of fermented orange peels.

Therefore, SSF resulted in an increase of CP at the expense of its availability for the rumen microorganisms. In the rumen, CP and amino acids can be degraded, deaminated and decarboxylated [52] to produce branched-chain volatile FA (BCVFA). The observed shift towards accumulation of these fermentation products in the fermented substrate would indicate that the reduction in availability would be compensated by the increase in the content. Moreover, BCVFA are known as essential nutrients for ruminal cellulolytic microorganisms [53], which agrees with the increased fiber content observed in the fermented substrate.

Chitin is a biopolymer structurally similar to cellulose, so it is not surprising that fungal growth resulted in an increase of the NDF proportion in the fermented substrate after SSF. The observed results agree with those reported by Joshi and Sandhu [54] and Oliveira, Feddern, Kupski, Cipolatti, Badiale-Furlong and de Souza-Soares [41], but disagree with those reported by Cooray and Chen [55], who found reductions in the NDF content after fermentation. This is not surprising because the effect depends on the nature of the fermented substrate. When the fermented substrate is a high lignified or fiber rich waste, the SSF contributes to degrade recalcitrant plant cell walls, reducing fiber content in the fermented residue [20,55], but when the initial substrate is not very lignified and rich in fiber, the growth of the fungal mycelium, rich in chitin, contributes to an increase in the fiber content of the obtained fermented residue [41,54].

SSF has been claimed to improve digestibility by reducing the levels of non-nutritive compounds that inhibit digestive enzymes (e.g., trypsin and chymotrypsin inhibitors) and promote protein crosslinking (e.g., phenolic and tannin compounds), as well as through the production of microbial proteases, which partially degrade and release some of the proteins [56,57]. The digestibility of the organic matter of vegetables, however, is also closely linked to that of the cell walls [58]. The extent of the degradation of the cell walls in the rumen depends essentially on the extent to which the walls are lignified [58]. Therefore, changes in the chemical composition due to fungal growth after SSF, such as the increased NDF and ADF contents, could also affect digestibility. As a consequence, when SSF was used with fruit and vegetable mix discards, a significant 27.2% reduction in IVOMD was observed. A similar decrease in the digestibility of potato processing waste [59], fermented orange peels and grape distillery stalks [51] and wheat bran [50] have been reported, which agree with the similar physicochemical characteristics of these substrates and the substrate used in the present study. However, although IVOMD decreased and fiber increased with the SSF process, fiber proportions of the solid state fermented fruit and vegetable discards, as well as digestibility values observed, are similar to those of good quality forages used in ruminant rations [60] and seemed to be appropriate to fulfill sheep nutritional requirements [61]. 

Gas and SCFA production are both directly related to the amount of organic matter fermented by rumen microorganisms [62]. In addition, as commented before, the SSF resulted in greater fiber contents that are less extensive and rapidly fermented by rumen microorganisms. Therefore, the lower values of SCFA and asymptotic gas production observed in the in vitro gas production trials for the fermented substrate compared to CTR would indicate that SSF substrate was fermented at a smaller extent than CTR and support the differences observed in IVOMD.

There were also differences in the SCFA profile due to the SSF process. Solid state fermentation of these wastes produced more acetate and less propionate with a concomitant greater acetate/propionate ratio, indicating a less efficient fermentation [63]. Differences in SCFA profile are again most likely related to the different carbohydrates’ composition [64] and agree with the increased fiber contents and reduced sugar content caused by fungal growth.

Fermentation of carbohydrates in the rumen provides energy for microbial growth. A high synthesis of microbial dry matter requires a high consumption of precursors necessary for microbial growth, which means that less of the fermentable substrate is available for production of SCFA [65]. In situations with a low efficiency of microbial synthesis, production of SCFA relative to TDS is increased; this is the case when SSF is used [65].

In order to avoid digestibility reduction with these wastes, further research is required to optimize the SSF process so that the fungal growth is achieved with a lower chitin creation. For instance, it has been reported that the type of microorganisms used as inoculum in the SSF, the nature of the solid substrates and the fermentation conditions are important parameters that influence the product yield and consequently affect the success of SSF process [66]. 

## 5. Conclusions

It can be concluded that SSF of fruit and vegetable discards succeeded in obtaining a stabilized raw material enriched in protein, but at the expense of a reduction of sugar content and an increase in NDICP and fiber, which, in turn, reduced its in vitro digestibility and led to a less efficient fermentation process. These changes in the nutritional profile of the fermented products should be taken into account before including them in ruminant’s rations.

## Figures and Tables

**Table 1 animals-11-01653-t001:** Effect of solid state fermentation process on the chemical composition of fruit and vegetable mix discards.

Item (g kg^−1^ DM Unless Otherwise Stated)	Treatment		SEM	*p*-Value
	SSF	CTR		
DM (g kg^−1^)	901	891	22	0.609
OM	940	960	0.5	<0.001
Reducing sugars	43	265	2.1	<0.001
CP	157	100	1.4	<0.001
NDF	398	196	20.1	<0.001
ADF	303	153	12.7	<0.001
NDICP (g kg^−1^ CP)	33	9.5	9.07	0.001

SSF: solid state fermentation, CTR: control (fruit and vegetable mix discards), DM: dry matter, OM: organic matter, CP: crude protein, NDF: neutral detergent fiber, ADF: acid detergent fiber, NDICP: neutral detergent insoluble protein, SEM: standard error of the mean.

**Table 2 animals-11-01653-t002:** Effects of solid state fermentation process on the in vitro gas production profile of fruit and vegetable mix discards.

Item	Treatment		SEM	*p*-Value
	SSF	CTR		
A (m Lg DM^−1^)	207	303	13.8	0.001
c (h^−1^)	0.057	0.061	0.0103	0.676
L (h)	1.10	0.75	0.687	0.574

SSF: Solid state fermentation, CTR: control (fruit and vegetable mix discards), A: gas production from the insoluble fraction, c: gas production rate constant for fraction A, L: lag time prior to gas production, DM: dry matter, SEM: standard error of the mean.

**Table 3 animals-11-01653-t003:** Effects of solid state fermentation process on in vitro digestibility and fermentation parameters of fruit and vegetable mix discards.

Item	Treatment		SEM	*p*-Value
	SSF	CTR		
IVOMD (g kg^−1^)	585	804	14.3	<0.001
SCFA (mmol L^−1^)	76.4	89.6	2.61	0.003
SCFA: TDS (mmol g OM^−1^)	308	260	6.6	<0.001
CH_4_ (mmol L^−1^)	24.8	29.3	0.81	0.002
CH_4_: TDS (mmol/g OM^−1^)	9.99	8.50	0.224	0.001
Individual SCFA proportions (mmol 100 mmol^−1^)				
Acetic	63.8	61.8	0.67	0.020
Propionic	20.1	20.9	0.17	0.004
Butyric	12.7	14.5	0.43	0.006
Isobutyric	0.702	0.472	0.0447	0.003
Valeric	1.38	1.51	0.090	0.155
Isovaleric	1.35	0.874	0.0887	0.003
Branched-chain FA	2.05	1.35	0.133	0.003
Acetic:propionic	3.19	2.96	0.055	0.008
(acetic+butyric):propionic	3.82	3.67	0.042	0.011

SSF: Solid state fermentation, CTR: control (fruit and vegetable mix discards), IVOMD: in vitro organic matter digestibility, SCFA: short chain fatty acid, OM: organic matter, FA: fatty acids, TDS: truly digestible substrate, SEM: standard error of the mean.

## Data Availability

The datasets generated during and/or analyzed during the current study are available from the corresponding author on reasonable request.

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
