# Peer review of "Solid State Fermentation as a Tool to Stabilize and Improve Nutritive Value of Fruit and Vegetable Discards: Effect on Nutritional Composition, In Vitro Ruminal Fermentation and Organic Matter Digestibility"

_animals, 2021, doi:10.3390/ani11061653_

Round 1

Reviewer 1 Report

General comment

The manuscript aimed to investigate the effect of solid state fermentation of fruit and vegetables on nutrient content and digestibility. The study is interesting because alternative to in feed proteins or new ingredients the do not compete with human nutrition are needed.

The manuscript needs few revisions, and some information are needed for better understand and describe the study. More comments are reported below:

Simple Summary

"in vitro" in Italic for the whole manuscript

Line 13-15: Please, format the manuscript.

Abstract

Line 25: Please specify the provenience of fruit and vegetables and the main chemical components (moisture or dry matter, protein, fat, ash etc..). Also add a brief description of the experiment, if there was a control group etc..

Introduction

 Line 46-49: Agreed with authors about this sentence; However, the new challenge for feed production is not only the increasing demand of animals' feed but also find alternatives to in feed proteins. I suggest spending few sentences about this.

Line 72-74: please revise this sentence.

Line 76: "ruminal rations" what authors means?

M&M

Line 138: Please define 500 mg of sample. In M&M section author did not well explained which samples were collected, please specify.

Line 147: in chemical analysis please define all samples.

Results

Table 3: Please better format the table

Discussion

Line 250- 252: this sentence seems not linked with the sentence reported before and below. it is difficult to understand, I suggest to better explain that a proper treatment of fruits and vegetables waste is needed for....

line 253: " rations" I suggest to use the term " nutrition in order to be more clear in the explanation

Line 250: please add the citation as: Silveira et al. (50) and Ranjanet al. (51). These two studies adopted different substrates (Rice Bran, Wheat Bran and de-oiled rice bran) compare the present study (fruit and vegetables). I suggest to add references that use similar substrates as fruits and vegetables.

Line 326: please better define "appropriate for ruminants" indicating what is the "optimum" or requirements for ruminant. I suggest to focus on ewes due to the used of ewes rumen fluid. However, authors could add information about cattle. I suggest to use NRC (2001) for adding these information about fiber requirement

Author Response

Response to Reviewer 1 Comments

We thank you for the decision of considering our manuscript after substantial changes for publishing in your journal. We also thank the reviewer for the comments and suggestions she or he has proposed. Below, we have included the responses as well as the amendments suggested.

General comment

The manuscript aimed to investigate the effect of solid state fermentation of fruit and vegetables on nutrient content and digestibility. The study is interesting because alternative to in feed proteins or new ingredients the do not compete with human nutrition are needed.

The manuscript needs few revisions, and some information are needed for better understand and describe the study. More comments are reported below:

Simple Summary

"in vitro" in Italic for the whole manuscript

The changes have been done in the whole manuscript.

Line 13-15: Please, format the manuscript.

The format has been adjusted.

Abstract

Line 25: Please specify the provenience of fruit and vegetables and the main chemical components (moisture or dry matter, protein, fat, ash etc..). Also add a brief description of the experiment, if there was a control group etc..

Information on mixture discards have been added in the abstract. We have not added information on the experiment or the chemical composition because there is a limitation of words in the abstract and we think that this information is provided in the material and methods section. We have added two sentences:

Lines 25-26: Mixtures were composed approximately by 28 % of citric fruits, 35 % of other fruits and 37 % of vegetables.

Lines 36-38: SSF succeeded in obtaining a stabilized material enriched in protein, but at the expense of a reduction of protein availability and organic matter digestibility. 

Introduction

 Line 46-49: Agreed with authors about this sentence; However, the new challenge for feed production is not only the increasing demand of animals' feed but also find alternatives to in feed proteins. I suggest spending few sentences about this.

We have added extra information regarding the increasing demand of animal feed and different protein sources which have a negative impact on the sector.

Lines 50-53: “In addition, the demand for animal feed is increasing and also the need for new protein sources. Imported soybean meal is the main protein source used in animal feeding in the EU, which increases the environmental impact on the sector.”

Line 72-74: please revise this sentence.

This sentence has been reworded as suggested.

Lines 80-82: “On the contrary, few studies have been carried out about the effect of SSF on ruminal digestibility of fruit and vegetables discards, being most of them focused only on the protein content increase and bioactive compounds release.”

Line 76: "ruminal rations" what authors means?

There was a mistake; it was “ruminant” instead of “ruminal”. We have amended it (Line 84).

M&M

Line 138: Please define 500 mg of sample. In M&M section author did not well explained which samples were collected, please specify.

We have provided more information on the incubated samples, both in the short-term and long-term in vitro batch fermentation trials. 

Lines 143-144 and 155-156: “Approximately 500 mg of CTR or SSF sample of the three independent processing runs were weighed”.

Line 147: in chemical analysis please define all samples.

We have provided more information for clearness about the samples.

Line 165:” CTR and SSF samples of the three independent processing runs were dried.”

Results

Table 3: Please better format the table

We have tried to better format the table taking into account journal specifications.

Discussion

Line 250- 252: this sentence seems not linked with the sentence reported before and below. it is difficult to understand, I suggest to better explain that a proper treatment of fruits and vegetables waste is needed for....

The reviewer is right, we have deleted “which compels for its proper treatment”. In this way we think that sentences are better linked.

line 253: " rations" I suggest to use the term " nutrition in order to be more clear in the explanation

We have replaced “rations” with “nutrition” as suggested (Line 275).

Line 250: please add the citation as: Silveira et al. (50) and Ranjanet al. (51). These two studies adopted different substrates (Rice Bran, Wheat Bran and de-oiled rice bran) compare the present study (fruit and vegetables). I suggest to add references that use similar substrates as fruits and vegetables.

We have done the proposed changes in the references. We have also added a new reference related to orange peel in vitro ruminal digestibility.

Lines 313-314: “and Nicolini et al [51] who found a decrease in the in vitro true digestibility of fermented orange peels.

Nicolini, L.; Volpe, C.; Pezzotti, A.; Carilli, A. Changes in in-vitro digestibility of orange peels and distillery grape stalks after solid-state fermentation by higher fungi. Bioresour. Technol. 1993, 45, 17-20.

Line 326: please better define "appropriate for ruminants" indicating what is the "optimum" or requirements for ruminant. I suggest to focus on ewes due to the used of ewes rumen fluid. However, authors could add information about cattle. I suggest to use NRC (2001) for adding these information about fiber requirement

We have reworded this paragraph taking into account reviewer´s comments. We have added two new references.

Lines 352-354: “However, although IVOMD was decreased and fiber increased with the SSF process, fiber proportions of the solid state fermented fruit and vegetable discards, as well as digestibility values observed are similar to those of good quality forages used in ruminant rations [60] and seemed to be appropriate to fulfill sheep nutritional requirements [61]. “

  1. R., B.; Dulphy, J.P.; Sauvant, D.; Tran, G.; Meschy, F.; Aufrère, J.; Peyraud, J.L.; Champciaux, P. Les tables de la valeur des aliments In Alimentation des bovins, ovins et caprins. Besoins des animaux-Valeurs des aliments, Quae, É., Ed.; Versailles Cedex, France, 2007; pp. 181-275.
  2. Hassoun, P.; Bocquier, F. Alimentation des ovins In Alimentation des bovins, ovins et caprins. Besoins des animaux-Valeurs des aliments, Quae, É., Ed.; Versailles Cedex, France, 2007; pp. 121-136.

In addition, although none of the reviewers have pointed it out, with the aim of improving the manuscript we have included two additional references in the introduction section (Line 80):
-Rajesh, N.; Imelda, J.; Raj, R.P. Value addition of vegetable wastes by solid-state fermentation using Aspergillus niger for use in aquafeed industry. Waste Manag 2010, 30, 2223-2227, doi:https://doi.org/10.1016/j.wasman.2009.12.017.

-Sadh, P.K.; Saharan, P.; Duhan, J.S. Bio-augmentation of antioxidants and phenolic content of Lablab purpureus by solid state fermentation with GRAS filamentous fungi. Resource-Efficient Technologies 2017, 3, 285-292, doi:https://doi.org/10.1016/j.reffit.2016.12.007.”

These two references replace that of:

-Nicolini, L.; Volpe, C.; Pezzotti, A.; Carilli, A. Changes in in-vitro digestibility of orange peels and distillery grape stalks after solid-state fermentation by higher fungi. Bioresour. Technol. 1993, 45, 17-20.

In this way we think that it is more clearly referenced the previous sentence.

Reviewer 2 Report

The proposal is relevant, but the authors need to include some evaluations, mainly in the analysis of nitrogenous compounds associated with heating plant residues. Inclusion of fractions soluble N, N ammonia data are relevant. The way in which the vegetables were subjected to fermentation was not described in detail. It is assumed that the vegetables were kept in closed conditions in the absence of oxygen.

Author Response

Response to Reviewer 2 Comments

We thank you for the decision of considering our manuscript after substantial changes for publishing in your journal. We also thank the reviewer for the comments and suggestions she or he has proposed. Below, we have included the responses as well as the amendments suggested.

Comments and Suggestions for Authors

The proposal is relevant, but the authors need to include some evaluations, mainly in the analysis of nitrogenous compounds associated with heating plant residues. Inclusion of fractions soluble N, N ammonia data are relevant. The way in which the vegetables were subjected to fermentation was not described in detail. It is assumed that the vegetables were kept in closed conditions in the absence of oxygen.

An extra sentence has been added to the manuscript as an extra information.

Lines 131-133: Plastic trays were covered with lids but without closing them hermetically to enable the air entrance.

Animals-1181955

Solid state fermentation as a tool to stabilize and improve nutritional quality of fruit and vegetable discards: effect on nutritional composition, in vitro ruminal fermentation and organic matter digestibility

Title: Nutritional quality evaluation requires intake digestible nutrients and animal performance analyses. Present paper evaluated just chemical composition and in vitro digestibility, thus characterizing feed nutritional value. Suggested title: “Solid state fermentation as a tool to stabilize and improve nutritive value of fruit and vegetable discards: effect on nutritional composition, in vitro ruminal fermentation and organic matter digestibility”.

We agree with the reviewer that feedstuff evaluation require animal production trials in which intake and animal performance are evaluated. However, prior to that, information on the nutritional value of the feedstuff is necessary in order to formulate it in ration in combination with a concentrate in a total mixed ration.

The title has been changed following the proposed suggestion.

Line 53-55: “The major obstacles to their use as quality animal feed are the low nutritional value of some of these by-products (mainly due to low protein content), their high humidity and the possible presence of mycotoxins”: Include information regarding the non-fibrous and fibrous carbohydrate fractions of the vegetable residues submitted to fermentation process.

Information on non-fibrous (reducing sugars) and fibrous carbohydrate (NDF and ADF) fractions of the vegetable residues submitted to fermentation process (CTR) can be found in table 1. We consider that the introduction should not include information about the samples used in the study.

Line 56-57. “Therefore, it is essential to implement technologies to enhance their nutritional quality in a sustainable, economically feasible and controlled way”. Analysis of the literature related to fermentation effects on nutritional quality of vegetables shows a decrease in quality (Digestible nutrients intake) due to the transformation of non-fibrous carbohydrates in organic acids, as well some true protein in N ammonium. Therefore, it is not correct to report that the fermented product will have a higher quality than fresh vegetable residues.

We agree with the Reviewer. We have reworded the sentence to point out the objective of improving protein content.

Lines 63-64: “Therefore, it is essential to implement technologies to account for these aspects, specially to improve the protein content, in…”

Line 68-71: “During the fermentation of these substrates, fungi improve fermented products’ digestibility due to their ability to release several enzymes (amylases, xylases, celullases…) which degrade plant cell walls [22,23] allowing the rumen microorganisms to access the polysaccharides”. Review this phrase, amylase has no effect on fibrous fraction composition.

The sentence has been reworded by deleting amylases (Line 78).

Line 128. Inclusion information of rumen fluid donor and animal diet.

Rumen fluid donor and animal diet was already provided. 

Lines 135-142 : “rumen fluid was collected from one multiparous Latxa ewe slaughtered for production purposes. Before slaughtering ewes were fed for 3 weeks a basal diet (80 % meadow hay and 20 % compound feed) and had free access to fresh water and feed. Ruminal fluid was collected before the morning feeding, and strained through four layers of cheesecloth into a pre-warmed thermos flask.

Line 151-154: Neutral detergent fiber (NDF) was determined with use of an alpha amylase, but without sodium sulphite, and was expressed free of ash [32]”. Justify why using amylase to analyse the mixtures composed approximately by 28 % of citric fruits(tangerine, orange and lemon), 35 % of other fruits (nectarine, apple, pear, watermelon, pomegranate and banana) and 37 % of vegetables (tomato, pumpkin, onion, green bean, pepper, leek, artichoke, cabbage, carrot, broccoli, potato, asparagus, chard and lettuce).

 We used this protocol because it is the standard one we use in our lab. In addition potatoes, green bean, apple, banana, carrot,…) contain important amounts of starch.        

Line 154-155: “Neutral detergent insoluble protein (NDICP) was determined by analysing the NDF residues for Kjeldahl nitrogen”. Considering information of Line 112- 113: “Substrate was dried at 60 ºC for 2 hours and sterilized (121 ºC, 15 min) before inoculation”. It is important to observe the Maillard reaction considering soluble sugars content of vegetables residues. As well ADIN determination is necessary considering soluble sugar content and temperature (60 ºC) used to dry the samples.

There are two reasons why we have not performed the ADIN analysis. First, the ADIN fraction of feeds is usually assumed to be indigestible, and some feeding systems for ruminants use the ADIN analysis as a measure of nitrogen (N) availability. We have used NDICP as a measure of N availability, instead. Pearson correlation coefficient between NDICP and ADICP has been reported to be 0.81 (Ki et al 2017) with other forages. Therefore, the conclusion we achieve using the NDICP as a measure of N availability would be the same if we had used the ADIN as a measure of N availability. It can be seen that we acknowledge that the SSF process results in a N availability reduction which is likely what would be concluded if the reviewer is right with the Maillard reaction.  Second, according to Marcos et al (2018), current ADIN analysis overestimates the N associated to ADFom (ADIN fraction), and this oversestimation is greater in fibrous samples, such is the case of the samples of the current trial.

Ki KS, Park SB, Lim DH, Seo S. Evaluation of the nutritional value of locally produced forage in Korea using chemical analysis and in vitro ruminal fermentation. Asian-Australas J Anim Sci. 2017;30(3):355-362. doi:10.5713/ajas.16.0626

C.N. Marcos, M.D. Carro, S. García, J. González. The acid detergent insoluble nitrogen (ADIN) analysis overestimates the amount of N associated to acid detergent fibre. Animal Feed Science and Technology. 2018. 244: 36-41

Table 01. Express NDICP values as a percentage of CP. Why the contents of ADIN and ammonia N were not been evaluated.

NDICP values have are now reported as percentage of CP. Why ADIN contents were not evaluated can be seen in the previous answer.

Line 292-293: “Therefore, SSF resulted in an increase in CP at the expense of its availability for the rumen microorganisms”. This is a key point in a data discussion considering the effects of nitrogen compounds on organic matter digestibility. Chemical composition of cell wall (NDF) affected degradation of the NDIN fraction. Considering the vegetables that were evaluated, it can be assumed that the lignin content is low, than NDF digestibility is high. However, we reiterate the need to evaluate the levels of ADIN (Acid insoluble Nitrogen) that can be produced from the Maillard reaction when the forage rich in reducing sugars and protein were dried at temperatures above 55 ºC.

We have used NDICP as a measure of N availability, instead. As stated before the Pearson correlation coefficient between NDICP and ADICP has been reported to be 0.81 (Ki et al 2017) with other forages. Therefore, greater ADIN values would be expected in the SSF samples due to a Maillard reaction leading to conclude that the CP increase would have been achieved at the expense of its availability, which is the same conclusion we find with the NDICP. We think that although determining the ADIN content would be interesting the conclusions of the current work are valid without it because we have measured the N availability with NDICP.

Line 353: “It can be concluded that SSF of fruit and vegetable discards succeeded in obtaining a stabilized row material enriched in protein, but at the expense of a reduction of sugar content and an increase in fiber which, in turn, reduced its digestibility and led to a less efficient fermentation process. These changes in the nutritional profile of the fermented products should be taken into account before including them in ruminant’s rations”. I suggest inclusion information associate to lignin and ADIN content of SSF of fruit and vegetable.

The conclusion has been reworded to include information on NDICP as a measure of N availability.

Lines 385 “content and an increase in NDICP and fiber which, in turn, reduced its in vitro digestibility”.

In addition, although none of the reviewers have pointed it out, with the aim of improving the manuscript we have included two additional references in the introduction section (Line 80):

-Rajesh, N.; Imelda, J.; Raj, R.P. Value addition of vegetable wastes by solid-state fermentation using Aspergillus niger for use in aquafeed industry. Waste Manag 2010, 30, 2223-2227, doi:https://doi.org/10.1016/j.wasman.2009.12.017.

-Sadh, P.K.; Saharan, P.; Duhan, J.S. Bio-augmentation of antioxidants and phenolic content of Lablab purpureus by solid state fermentation with GRAS filamentous fungi. Resource-Efficient Technologies 2017, 3, 285-292, doi:https://doi.org/10.1016/j.reffit.2016.12.007.”

These two references replace that of:

-Nicolini, L.; Volpe, C.; Pezzotti, A.; Carilli, A. Changes in in-vitro digestibility of orange peels and distillery grape stalks after solid-state fermentation by higher fungi. Bioresour. Technol. 1993, 45, 17-20.

In this way we think that it is more clearly referenced the previous sentence.

Round 2

Reviewer 2 Report

The authors partially answered our suggestions. It is necessary to discuss the data according to the results evaluated regarding the changes observed in the experiment and not only supported by data from the literature referring to the fungi chemical composition.

Author Response

Response to Reviewer 2 Comments

We thank you for the decision of considering our manuscript after substantial changes for publishing in your journal. We also thank the reviewer for the comments and suggestions she or he has proposed. Below, we have included the responses as well as the amendments suggested.

Comments and Suggestions for Authors

Line 151-154: Neutral detergent fiber (NDF) was determined with use of an alpha amylase, but without sodium sulphite, and was expressed free of ash [32]”. Justify why using amylase to analyse the mixtures composed approximately by 28 % of citric fruits(tangerine, orange and lemon), 35 % of other fruits (nectarine, apple, pear, watermelon, pomegranate and banana) and 37 % of vegetables (tomato, pumpkin, onion, green bean, pepper, leek, artichoke, cabbage, carrot, broccoli, potato, asparagus, chard and lettuce).

As we mentioned in our previous response to reviewer’s suggestion we used this protocol because it is the standard one we use in our lab. In addition, in our opinion it is suitable for our samples because several ingredients of the vegetable and fruit discards (potatoes, green bean, apple, banana, carrot…) contain important amounts of starch. Moreover, we do not think it is necessary to include this justification in the discussion of the results. Anyway, if the reviewer thinks that this answer is not good enough we would appreciate very much if he/she could argue why not.    

Line 154-155: “Neutral detergent insoluble protein (NDICP) was determined by analysing the NDF residues for Kjeldahl nitrogen”. Considering information of Line 112- 113: “Substrate was dried at 60 ºC for 2 hours and sterilized (121 ºC, 15 min) before inoculation”. It is important to observe the Maillard reaction considering soluble sugars content of vegetables residues. As well ADIN determination is necessary considering soluble sugar content and temperature (60 ºC) used to dry the samples.

Regarding the Maillard reactions, it is important to highlight that both subsamples (CTR and SSF) were subjected to the drying procedure (60 °C). Therefore, if a Maillard reaction occurred it would have happened to both of them (CTR and SSF). Therefore, any difference observed in NDICP could not be linked to a Maillard reaction because it is likely that this process, as mentioned above, would have affected both subsamples. Anyway we have reworded these lines to clearly indicate that both samples were dried following the same procedures.

Line 128: Both subsamples were dried at 60 °C for 2 hours.

Anyway, and as mentioned in the previous response there are two reasons why we have not performed the ADIN analysis. First, the ADIN fraction of feeds is usually assumed to be indigestible, and some feeding systems for ruminants use the ADIN analysis as a measure of nitrogen (N) availability. We have used NDICP as a measure of N availability, instead. Pearson correlation coefficient between NDICP and ADICP has been reported to be 0.81 (Ki et al 2017) with other forages. Therefore, the conclusion we achieve using the NDICP as a measure of N availability would be the same if we had used the ADIN as a measure of N availability. It can be seen that we acknowledge that the SSF process results in a N availability reduction. Second, according to Marcos et al (2018), current ADIN analysis overestimates the N associated to ADFom (ADIN fraction), and this overestimation is greater in fibrous samples, such is the case of the samples of the current trial.

For all these reasons we have not performed ADIN analysis. Anyway, if the reviewer thinks that this answer is not good enough we would appreciate very much if he/she could argue why not. We would appreciate if Reviewer could give us specific suggestions with evidence from other sources or literature supporting why the response we provide is not suitable.        

Ki KS, Park SB, Lim DH, Seo S. Evaluation of the nutritional value of locally produced forage in Korea using chemical analysis and in vitro ruminal fermentation. Asian-Australas J Anim Sci. 2017;30(3):355-362. doi:10.5713/ajas.16.0626

C.N. Marcos, M.D. Carro, S. García, J. González. The acid detergent insoluble nitrogen (ADIN) analysis overestimates the amount of N associated to acid detergent fibre. Animal Feed Science and Technology. 2018. 244: 36-41

 Line 269-273: “It has been also shown that SSF is one of the most suitable and economic techniques for detoxifying or enhancing protein enrichment, as well as for an efficient digestion and utilization of lignocellulosic agricultural fibrous feeds and fodder residues. Therefore, this can enhance their feed values [40], and bring benefits both to the economy and the environment, promoting the circular economy”. Analysis of the data in Tables 01, 03 shows a significant decrease in the content of reducing sugars, proportional increase in the values of NDF, ADF and CP.

We have reworded this paragraph to show the benefits of SSF reported in the literature. In our opinion, including the references it is clearly assessed that the previous statement was not based in our results.

Lines 301-307: In this sense, some authors report SSF as a promising alternative in the use of these agro-industrial by-products as a culture medium in order to account for these problems, making its use feasible in animal feed [40]. It has been also shown that SSF is one of the most suitable and economic techniques for detoxifying or enhancing protein enrichment, as well as for an efficient digestion and utilization of lignocellulosic agricultural fibrous feeds and fodder residues [14,18]. Therefore, this can enhance their feed values and bring benefits both to the economy and the environment, promoting the circular economy.

Lately in the discussion section we discuss the mentioned observed results in Table 1 and 3.

What are the justifications for the increase in CP levels in relation to fungi growth. Insert data related to the chemical composition of the fungus that support these values.

It is not possible to divide the fungus mycelium from the substrate after the SSF process for the analysis of the fungus mycelium.

As illustrated by López-Gómez et al (2020), in the figure 7.1D a filamentous fungi is growing on the surface of the solids. As they explained “the mycelium can penetrate deeply into the solid substrate particles, consuming nutrients and oxygen while excreting metabolites and enzymes”.

Some data about the theoretical composition of the fungus mycelium is provided in line 314-321 justifying the observed CP content change.

López-Gómez, J.P.; Manan, M.A.; Webb, C. Chapter 7 - Solid-state fermentation of food industry wastes. In Food Industry Wastes (Second Edition), Kosseva, M.R., Webb, C., Eds.; Academic Press: 2020; pp. 135-161.

The data show an increase in the CP fraction, NDIN as well as fiber associated to the decrease in the carbohydrate soluble fraction of vegetable residues. It is important to observe expressive decrease in IOMD, proving the inefficiency of the system in terms of preserving the nutritive value of stored products.

We agree with the reviewer that any conserved feedstuff (silage, hay…) implies a nutritive value reduction compared to the fresh initial feedstuff. As such in line 373-385 we discuss the implications of the observed effects on digestibility and its relationship with the changes on chemical composition. However, we do not agree with the reviewer, in that it is an inefficient system to preserve the nutritive value of stored products when the conserved feedstuff still has a IVOMD of 58% with a CP of 15,8%. These values suggest that SSF can provide stabilized conserved feedstuff suitable to be used in ruminant rations.

Line 292-293: “Therefore, SSF resulted in an increase in CP at the expense of its availability for the rumen microorganisms”. This is a key point in a data discussion considering the effects of nitrogen compounds on organic matter digestibility. Chemical composition of cell wall (NDF) affected degradation of the NDIN fraction. Considering the vegetables that were evaluated, it can be assumed that the lignin content is low, than NDF digestibility is high. However, we reiterate the need to evaluate the levels of ADIN (Acid insoluble Nitrogen) that can be produced from the Maillard reaction when the forage rich in reducing sugars and protein were dried at temperatures above 55 ºC.

Regarding the Maillard reactions, it is important to highlight that both subsamples (CTR and SSF) were subjected to the drying procedure (60 °C). Therefore, if a Maillard reaction occurred it would have happened to both of them (CTR and SSF). As a consequence any difference observed in NDICP could not be linked to a Maillard reaction because it is likely that this process, as mentioned above, would have affected both subsamples. Anyway we have reworded these lines to clearly indicate that both samples were dried following the same procedures.

Line 128: Both subsamples were dried at 60 °C for 2 hours.

As we have stated in the previous revision, we have used NDICP as a measure of N availability, instead. As stated before the Pearson correlation coefficient between NDICP and ADICP has been reported to be 0.81 (Ki et al 2017) with other forages. We think that although determining the ADIN content would be interesting the conclusions of the current work are valid without it because we have measured the N availability with NDICP.

Anyway, if the reviewer thinks that this answer is not good enough we would appreciate very much if he/she could argue why not. We would also appreciate if Reviewer could give us specific suggestions with evidence from other sources or literature supporting why the response we provide is not suitable.            

Line 305-310: “This is not surprising, because the effect depends on the nature of the fermented substrate. When the fermented substrate is a high lignified or fiber rich waste the SSF contributes to degrade recalcitrant plant cell walls, reducing fiber content in the fermented residue [20,55], but when the initial substrate is not very lignified and rich in fiber, the growth of the fungal mycelium, rich in chitin, contributes to an increase in the fiber content of the obtained fermented residue [42,54]”. Data from table 01 show expressive decrease in the fraction of sugars and NDF, ADF values doubled in the SSF. Considering that fungi growth of fungi can contribute to the SSF cell wall values, please insert data of chemical composition and fungi mass observed in treated vegetable residue.

It is not possible to divide the fungus mycelium from the substrate after the SSF process for the analysis of the fungus mycelium. As explained above, by López-Gómez et al (2020), illustrated in the figure 7.1D a filamentous fungi is growing on the surface of the solids. As they explained “the mycelium can penetrate deeply into the solid substrate particles, consuming nutrients and oxygen while excreting metabolites and enzymes”.

Some data about the theoretical composition of the fungus mycelium is provided in line 314-321 justifying the observed CP content change.

López-Gómez, J.P.; Manan, M.A.; Webb, C. Chapter 7 - Solid-state fermentation of food industry wastes. In Food Industry Wastes (Second Edition), Kosseva, M.R., Webb, C., Eds.; Academic Press: 2020; pp. 135-161.

Some data about the theoretical composition of the fungus mycelium is provided in line 325-328 justifying the observed NDF content change.

Line 353: “It can be concluded that SSF of fruit and vegetable discards succeeded in obtaining a stabilized row material enriched in protein, but at the expense of a reduction of sugar content and an increase in fiber which, in turn, reduced its digestibility and led to a less efficient fermentation process. These changes in the nutritional profile of the fermented products should be taken into account before including them in ruminant’s rations”. I suggest inclusion information associate to lignin and ADIN content of SSF of fruit and vegetable.

The conclusion has been reworded to include information on NDICP as a measure of N availability. Lignin is a very good indicator of IVOMD. The reviewer should take into account that we provide the actual value of IVOMD so the importance of the lignin value is less in our case.

Lines 417 “content and an increase in NDICP and fiber which, in turn, reduced its in vitro digestibility”.